# Small Extracellular Vesicles Derived from Induced Pluripotent Stem Cells in the Treatment of Myocardial Injury

**DOI:** 10.3390/ijms24054577

**Published:** 2023-02-26

**Authors:** Wan-Ting Meng, Hai-Dong Guo

**Affiliations:** 1Academy of Integrative Medicine, Shanghai University of Traditional Chinese Medicine, Shanghai 201203, China; 2Department of Anatomy, School of Basic Medicine, Shanghai University of Traditional Chinese Medicine, Shanghai 201203, China

**Keywords:** induced pluripotent stem cells, extracellular vesicles, exosome, myocardial injury, heart, mechanisms

## Abstract

Induced pluripotent stem cell (iPSC) therapy brings great hope to the treatment of myocardial injuries, while extracellular vesicles may be one of the main mechanisms of its action. iPSC-derived small extracellular vesicles (iPSCs-sEVs) can carry genetic and proteinaceous substances and mediate the interaction between iPSCs and target cells. In recent years, more and more studies have focused on the therapeutic effect of iPSCs-sEVs in myocardial injury. IPSCs-sEVs may be a new cell-free-based treatment for myocardial injury, including myocardial infarction, myocardial ischemia–reperfusion injury, coronary heart disease, and heart failure. In the current research on myocardial injury, the extraction of sEVs from mesenchymal stem cells induced by iPSCs was widely used. Isolation methods of iPSCs-sEVs for the treatment of myocardial injury include ultracentrifugation, isodensity gradient centrifugation, and size exclusion chromatography. Tail vein injection and intraductal administration are the most widely used routes of iPSCs-sEV administration. The characteristics of sEVs derived from iPSCs which were induced from different species and organs, including fibroblasts and bone marrow, were further compared. In addition, the beneficial genes of iPSC can be regulated through CRISPR/Cas9 to change the composition of sEVs and improve the abundance and expression diversity of them. This review focused on the strategies and mechanisms of iPSCs-sEVs in the treatment of myocardial injury, which provides a reference for future research and the application of iPSCs-sEVs.

## 1. Introduction

Cardiovascular disease (CVD) is the leading cause of global morbidity and mortality [1,2], with a 50% increase in associated mortality over the last 30 years [3]. In view of the heavy social burden, there is an urgent need for effective prevention and control measures. At present, surgery and drugs are the standard methods for the treatment of CVD, but they cannot promote the regeneration of damaged myocardial tissue [4]. The myocardial injury caused by a large number of cardiomyocyte apoptoses is irreversible [5]. Induced pluripotent stem cells (iPSCs) are reprogrammed cells that have features similar to embryonic stem cells, such as self-regeneration without restriction and differentiation into different tissue or cell types [6,7,8]. Compared with embryonic stem cells, they have abundant sources and have no ethical issues. Moreover, iPSCs induced by autologous cells can also reduce the risk of immune rejection and can be used as a potential treatment for CVD [9,10].

Like other cell therapies, iPSCs also have disadvantages such as low cell survival, retention and implantation rates of cells [11]. Recently, many studies have confirmed that stem cells play a therapeutic role in CVD mainly by inducing paracrine/autocrine growth factors, immunomodulators, and other bioactive molecules stored in their extracellular vesicles (EVs) [12,13,14]. EVs can be classified into apoptotic bodies (50~1000 nm in diameter), microvesicles (MVs) (100~1000 nm), and exosomes (40~160 nm, average ~100 nm) based on their origin [15]. With respect to the biogenesis of EVs, apoptotic bodies are released by dying cells, which are seldomly used for study possibly due to their large and uneven particle size. MVs are formed by the direct outward budding of plasma membranes. The specific process of exosome biogenesis is recognized as a “swallow and spit” process [16] (Figure 1A). Given that the latest MISEV guidelines suggest the use of “EVs” to generally denote a heterogeneous extracellular vesicle population, and “exosomes” are defined as small extracellular vesicles (sEVs), in this review, we focus on exosomes.

In the myocardial infarction (MI), myocardial ischemia–reperfusion injury (MIRI), and heart failure (HF) models, studies using stem cell EVs have shown that they can improve cardiac contractile function in the long term by reducing the initial infarct size, promoting angiogenesis, reducing fibrosis, and remodeling [17]. EVs derived from stem cells regulate gene expression by transferring different substances (including protein, DNA, mRNA, microRNA (miRNA), long-stranded non-coding RNA (lncRNA), and circular RNA (circRNA)) to achieve targeted regulation between cells [18,19], and they have the advantages of high biocompatibility, circulatory stability, and low immunogenicity [20], which open up a new field for resolving the obstacles of stem cell therapy (Figure 1B).

IPSC-derived extracellular vesicles (iPSCs-EVs) can play a therapeutic role similar to that of iPSCs, and iPSCs-EVs are easier to store and transport [21]. At the same time, some limitations of cell therapy, such as embolism and tumor occurrence, are avoided [22,23]. According to the comparison of EVs secreted from mesenchymal stem cells (MSCs) and iPSCs, it was found that while iPSC-EVs enclose proteins that modulate RNA and microRNA stability and protein sorting, MSC-derived EVs are rich in proteins that organize the extracellular matrix, regulate locomotion, and influence cell–substrate adhesion. Moreover, compared to their respective cells, iPSC-EVs share 76.63% of proteins with iPSCs [24], including proteins involved in angiogenesis signaling pathways (VEGF, TGFB1, and Angiogenin) [25], proteins related to membrane organization and the wound-healing process (HSPA5, RAB10, and CLIC1) [26], and proteins involved in cardiac development and cardiac mechanical and electrical function (GSTM, ARGBP2, CDH11, and ACTA2) [27].

## 2. Isolation of sEVs from Induced Pluripotent Stem Cells

The efficacious extraction of iPSC-derived extracellular vesicles (iPSC-sEVs) is a prerequisite for them to play a therapeutic role. How the high yield, high purity, and high biological activity of small extracellular vesicles can be obtained is directly related to future research and applications [28,29]. At present, many techniques for isolating sEVs have been developed, which depend to a large extent on the physical and chemical properties of sEVs, and the choice of methods should also take into account specific research needs. The isolation methods of iPSC-sEVs for the treatment of CVD include ultracentrifugation (UCF) [30,31], size-exclusion chromatography (SEC) [32], polymer-based precipitation [33], affinity capture [34], magnetic [35,36] and anion-exchange-based methods [37], or a combination of the aforementioned methods [38]. In this mini review, we will introduce three of the most common ones in detail.

UCF is the “gold standard” for isolating sEVs and the most commonly used technology [39]. The substances with different densities and sizes are separated by using different centrifugal forces and velocities (Figure 2A). First, larger cells, cell debris, and dead cells are removed by low-speed centrifugation [40]; then, resuspension with PBS is performed, and finally, ultracentrifugation is carried out to remove contaminated proteins to obtain granular exosomes [41]. The temperature of the whole centrifugation process is kept at 4 °C to ensure that proteases, DNA enzymes, and ribonucleases are inactivated [42]. The concentration of exosomes is determined using an enhanced BCA protein analysis kit [43,44] or nanoparticle tracking analysis, which is an optical particle tracking method developed to determine the concentration and size distribution of particles [45]. In addition, Western blotting can provide useful information on the size of the different proteins [46]. ELISA is another established technique for protein quantification and could be executed in multiple different assay formats [47]. Unlike Western blotting and ELISA, which quantify targeted proteins on a relatively small scale, mass spectrometry enables high-throughput peptide profiling [48]. Additionally, small EVs can be characterized by observation under a transmission electron microscope (TEM) [49]. Moreover, a TEM can also be coupled with immunogold labeling (immuno-EM) to provide molecular characterization [50]. UCF has advantages of simple operation, low cost, and repeatability, and it is suitable for large volume samples [51]. Dong et al. [52] found that when exosomes were separated from plasma, UCF had the highest separation purity.

However, UCF is time-consuming, and different individual operations will also lead to different results [53]. In particular, repeated ultra-high-speed centrifugation has adverse effects on the quality and quantity of exosomes [54,55]. Their structural and biological integrity may also be damaged [56]. The appearance of the isodensity gradient centrifugation method is an improvement of UCF. By constructing a density gradient medium (gradually increasing from the top to the bottom of the centrifuge tube), exosomes and the corresponding isodensity area settle together under the effect of centrifugal force, thus removing most of the contaminated proteins [57] (Figure 2B).

SEC is a widely recognized method that uses polymers to form porous stationary phases in chromatographic columns. Exosomes are separated according to the different path lengths of molecules or particles with different sizes [58] (Figure 2C). Compared with UCF, the exosomes separated via SEC have more complete physical structures and biological functions [59] and are suitable for various biological fluids [60]. However, the products obtained via the SEC method may be contaminated by a large number of proteins with low purity, which means the method is suitable for samples with small size and high yield.

## 3. Drug Delivery of iPSC-sEVs in the Repair of Myocardial Injury

EVs can transfer encapsulated proteins and genetic information to recipient cells and act as information messengers between cells [61]. They are natural biologics with autologous origin, while they also maintain cargo integrity and stability. Furthermore, exosomal membranes contain certain proteins that have binding affinities to specific receptors on the surface of the recipient cells. EV uptake may occur through three mechanisms: endocytosis, ligand–receptor uptake, and fusion [62]. Upon binding to a specific target cell, EVs have the ability to initiate intracellular signaling via receptor–ligand interactions, undergo internalization via endocytosis and/or phagocytosis, or even fuse with the target cell’s membrane, resulting in the transfer of their contents to the cytosol of the recipient cell. These processes ultimately lead to the modification of the physiological state of the recipient cell [63].

Rab GTP enzymes such as Rab11, Rab35, Rab27a, and Rab27b participate in the production of exosomes through vesicle budding [64,65,66]. The expression of exosomal markers such as CD63 was shown to be reduced by the silencing of Rab27a and Rab27b [67,68]. To demonstrate in vivo EV transfer between cells, a few groups have recently developed clever modifications of EVs, allowing their behavior and target cells to be tracked in vivo. For example, Lai et al. combined Gaussia luciferase with metabolic biotinylation to create a sensitive EV reporter for multimode imaging, showing that the dynamic processing of EVs has an accurate spatio-temporal resolution [69]. In order to further evaluate the accuracy of time and space, Lai et al. also designed optical reporters to label multiple EV populations, and they found that EV-borne mRNA transfer between cells and the process is dynamic and multidirectional [70]. IPSC-sEVs contain mRNAs which participate in a variety of biological processes of cell proliferation, promoting angiogenesis and paracrine response [71]. In addition to proteins and mRNAs, miRNAs and other non-coding RNAs are also possible active EVs cargoes. The miRNA secreted in sEVs can be functionally delivered to target cells, resulting in the direct modulation of their mRNA targets [72]. Mendel et al. reported miRNAs to be present in both iPSCs and iPSC-sEVs; they found miR-19b, miR-20a, miR-126-3p, miR-130a-3p and miR-210-3p were reportedly involved in the promotion of angiogenesis, adaptation to hypoxic stress, and regulation of cell cycles [73].

Exosomes are also highly engineerable, and the strategies include genetic engineering and chemical modification [74,75]. The engineering of exosomal surface proteins confers cell and tissue specificity [76]. The surface molecules anchored on exosomes from different cell sources vary, which endows them with selectivity for specific recipient cells. Bobis-Wozowicz et al. showed that iPSC-sEVs are able to transfer bioactive molecules delivered to human cardiac mesenchymal stromal cells and were found to exert protective effects by affecting the transcriptomes and proteomic profiles of the recipient cells [77]. Additionally, iPSC-sEVs combined with small-molecule RNA (miR-499) induce myocardial differentiation and improve cardiac function through the wnt/β-catenin signaling pathway in rats [78]. Jung et al. found that exosomal cargo containing miR-106a-363 improved the murine LV ejection fraction and reduced the myocardial fibrosis of the injured myocardium [79]. For the application of an in vivo model, 15–100 μg is the commonly used dose for the treatment of mouse or rat models [44], while 2–40 μg/mL is the commonly used intervention dose for in vitro studies [80].

Consequently, EVs from gene-edited patient-specific iPSCs can be directed to the specific lesions of each individual patient to promote the salvage of the existing injured cells. IPSC-sEVs hold potential for a wide spectrum of beneficial effects on cell function recovery to restore the myocardial injury by simulating and activating the endogenous repair, consisting of the native transfer of proteins, mRNAs, and miRNAs (Table 1). EVs represent the most feasible approach to translate the enormous potential of pluripotent stem cell biology.

## 4. Mechanism of iPSC-sEVs in the Repair of Myocardial Injury

Studies have found that iPSC-sEVs play a protective role in the treatment of CVD by regulating apoptosis, inflammation, and fibrosis, as well as promoting angiogenesis [94,95,96]. These are achieved through cell-to-cell communication, which is promoted by substances such as miRNA, small molecules, and proteins (Figure 3).

### 4.1. MI

The death of many CMs after MI leads to strong inflammation. IPSC-sEVs show angiogenesis and anti-inflammatory potential in the cell therapy of MI [97]. Angiogenesis is the main mechanism of improving left ventricular function through cell therapy after ischemic myocardial injury, which indicates that iPSC-sEVs are a potential target for MI therapy [98]. More and more studies have shown that exosomes derived from iPSCs can promote endogenous repair and enhance cardiac function after MI [79,99]. Takeda et al. [83] isolated exosomes from human iPSCs and administered them successively in the ischemic myocardial model of mice, which showed that iPSC-sEVs significantly improved myocardial injury after MI by reducing apoptosis and fibrosis. In vitro studies have also shown that angiogenesis and anti-apoptotic effects depend on the increased survival of CMs derived from iPSCs, and exosomes from iPSC-derived CMs (iPSC-CMs) improve myocardial recovery without increasing the probability of arrhythmogenic complications [100]. Gao et al. [101] demonstrated that exosomes from human iPSC-CMs also have cardioprotective effects in a swine MI model according to the ejection fraction, wall stress, myocardial bioenergetics, and cardiac hypertrophy. In vitro studies also showed their angiogenic and anti-apoptotic effects depending on increased endothelial cell tube formation and the survival of CMs derived from hiPSCs.

### 4.2. MIRI

IPSCs-sEVs can promote myocardial regeneration in MIRI, partly due to its ability to shuttle between cells, which contains a large amount of miRNA, especially miR-146a [102]. MiR-146a inhibits IRAK1 and TNF receptor-related factor 6 to reduce the activation of NF-κβ to increase cardiac function and reduce myocardial fibrosis after MIRI [103]. Furthermore, miR-21 has been proved to have beneficial effects on damaged myocardium [104,105]. MiR-21 reduces cardiomyocyte apoptosis by regulating the expression of PDCD4 and AKT pathways [106]. IPSC-sEVs are also involved in regulating signaling pathways such as WNT [107], which partially remuscularize the injured region, restore cardiac function, and reduce fibrosis in the infarcted hearts of rats by regulating actin cytoskeleton and immunogenicity. IPSC-sEVs have anti-apoptotic and antioxidant effects [108]. For example, iPSC-EVs can protect H9c2 cells from H_2_O_2_-induced oxidative stress by inhibiting the activation of caspase3/7. The intramyocardial injection of iPSC-sEVs before reperfusion can protect against MIRI. Furthermore, IPSC-sEVs deliver cardioprotective miRNAs, including nanog-regulated miR-21 and HIF-1α-regulated miR-210 [82].

### 4.3. Coronary Heart Disease

Coronary heart disease (CAD) is caused by coronary artery stenosis or obstruction due to atherosclerosis. According to the current view, oxidative stress, endothelial dysfunction, and inflammation are the three key factors for the occurrence and development of CAD [109,110]. Many studies have focused on the use of natural drugs and biodegradable synthetic materials for scaffolds. However, recent studies have combined the use of EVs derived from iPSCs, providing a promising solution for vascular tissue engineering [111]. IPSC-sEVs participate in paracrine and autocrine communication between cardiovascular cells through miRNAs and other mediators [112]. EVs released from iPSCs have been shown to have myocardial protective effects, which can improve the survival rate of CMs. This process is achieved by inducing macrophage polarization and reducing the transcription level of protein kinase by miR-181b [113]. Wang et al. [114] pointed out that iPSC-sEVs can increase type III collagen and fibronectin, increase vascular permeability, optimize the vascular environment, and improve cardiac function. More and more studies have confirmed that EVs from mesenchymal stem cells (MSCs) are effective drug carriers for the treatment of CAD, but their application is hindered by donor variation and traditional tissue-derived MSC expansion limitations [102,115]. While small EVs prepared from standardized MSCs derived from iPSCs (iMSC-sEVs) have unlimited scalability and have the ability to target CAD therapy [116], some studies show that they have a better protein structure than iPSC-sEVs, providing more possibilities for the prevention and treatment of CAD [117,118].

### 4.4. HF

The lost myocardium after MI is usually replaced by non-contractile scar tissue, which can lead to congestive heart failure (HF). As CMs are terminally differentiated cells with minimal regenerative capacity, heart transplantation is the gold standard for the treatment of HF, which faces the obstacles of the shortage of donor hearts, complications after transplantation, and the long-term failure of the transplanted heart [119]. Tian et al. [120] reviewed that the regulation of miRNAs rich in iPSCs-sEVs on Nrf2 and antioxidant proteins in the heart and brain mediates cardiac function and sympathetic excitation during HF. It is speculated that the targeted uptake ability of receptor cells can be increased when engineering exosomes with specific miRNAs or antagomirs is used to treat HF. Qiao et al. [121] confirmed that iPSC-sEVs alleviate cardiac dysfunction by regulating the Akt pathway through miR-21-5p. In recent years, lncRNA has become a key regulator of biological processes involved in the progression of HF [122]. Viereck et al. [123] focused on the potential of highly conservative lncRNAH19 and found that its expression was down-regulated in HF. The iMSC-sEVs also play an important role in heart failure. Hou et al. [124] found that iMSC-sEVs protected endothelial cells from oxidative stress by activating the Akt/Nrf2/HO-1 signaling pathway in HF models.

## 5. Challenges in the Treatment of CVD with IPSC-sEVs

It has been confirmed that iPSC-sEVs promote heart repair after MI, which means they are superior to iPSCs [21]. EVs provide a feasible alternative cell-free therapy in iPSC medicine. Because of their low immunogenicity, they does not seek a host immune response, so there is no need to match donor and recipient [125,126]. However, there are still many problems with the treatment of EVs, such as their production, stability, half-life, and delivery efficiency. Therefore, it is particularly necessary to comprehensively analyze the chemical and functional characteristics of the EVs and to study their physiological characteristics, diversity, and transport mode.

Chandy et al. [127] drew a map of microRNAs in cardiac extracellular secretions derived from human iPSCs. Human iPSCs were differentiated into iPSC-CMs, iPSC-ECs and iPSC-CFs, and the EVs were isolated. Their miRNA content was sequenced and compared with the source cells. Interestingly, only a part of cells miRNAs was found to be secreted in the EVs and was cell-specific. A comparative analysis showed a decrease in miR-22 expression in exosomes from cardiac-fibroblast-derived hiPSCs compared with dermal-fibroblast-derived hiPSC exosomes [27]. Future research needs to conduct in-depth sequencing analyses to understand the role of other non-coding RNAs in mediating the improvement of cardiac function. In addition, since iPSC-sEVs carry miRNA and each miRNA has multiple target genes, it is also necessary to prevent the occurrence of adverse non-target effects.

IPSCs differentiate into CMs, which are equivalent to fetal CMs, and lack the electrophysiological and ultrastructural characteristics of mature CMs [128,129], such as fully functional seromuscular reticular structure and transverse canal system. After differentiation, the maximum contractility was lower, calcium storage and circulation decreased, and the mitochondrial function was immature [130]. In addition, the EVs’ function was also affected. The current research is mainly focused on using the paracrine function of iPSCs to play a role, rather than ensuring they differentiate into therapeutic cells [131,132,133]. Even so, the content and level of iPSC-sEVs will change after serum starvation and hypoxia treatment [134,135], which makes clinical treatment more difficult. Nachlas et al. [136] highlight the importance of a 3D culture environment to influence cell phenotype and function. In addition, 3D-printed cardiac patches and personalized hydrogel can help iPSCs’ further maturation [136,137,138]. Furthermore, gene editing technology can be used to achieve the richness of iPSC cells [26,139]. For instance, CRISPR/Cas9 is used for gene editing based on homologous recombination to obtain mutation-corrected iPSCs so that the pathogenic mutation can be corrected without preserving the genetic footprint [26].

However, if these molecules are to be used in clinical therapy, the standard procedures for purifying exosomes need to be optimized. Overall, small EVs play critical roles in cell–cell communication through endocytosis, phagocytosis, and membrane fusion. EV uptake was found to correlate with intracellular and microenvironmental acidity [140,141], suggesting that the microenvironment influences the delivery efficiency of EVs. In the case of factors operating at the intracellular level, delivery into the correct cellular compartments while maintaining the stability, integrity, and biological potency of these factors remains challenging.

Furthermore, the content of exosomes can be modified by stress preconditioning [142], serum deprivation [143], or the genetic modification and epigenetic reprogramming of iPSCs [144,145,146]. Recent studies show that exosomes can cross the BBB (blood–brain barrier), and a leaky BBB state in mental disorders (such as stress, depression, and schizophrenia) may be initiated by exosomes released from cells being influenced by this disease state [147]. Chronic stress can cause immune disorders and inflammatory responses. Moreover, exosomal components are strongly influenced by inflammatory signals such as LPS, tumor necrosis factor (TNF)-α [148], and interferon (IFN)-γ [149]. They could modulate the therapeutic efficacy via the regulation of differential gene expressions [150,151] and largely influence the effect of iPSC-sEV treatment.

## 6. Prospects and Conclusions

The potential of IPSC-sEVs in the treatment of CVD is exciting. Compared with cells, EVs cannot self-replicate, which reduces tumor toxicity. The future application of IPSC-sEVs is likely to be combined with other drugs or systems. With the development of front-line technologies, including scRNA-seq, multi-omics, genome editing, and machine learning, they possess great potential for the analysis of exosome contents and their transfer specificity [152]. Exosomes can be endogenously modified by the genetic modification of production cells to produce cells overexpressing desired therapeutic substances that are eventually incorporated into exosomes upon secretion [153]. Alternatively, exosomes can be loaded exogenously using various techniques, such as sonication [154], membrane permeabilization [155], and extrusion [156]. Therefore, in order to ensure their safety and effectiveness, a number of challenges must be addressed, including the characteristics of the content, specific molecular mechanisms for disease treatment, and biosafety as a drug delivery system. In short, more basic research and new technologies are needed to fully realize the therapeutic potential of exosomes derived from iPSCs and accelerate their clinical application.

## Figures and Tables

**Figure 1 ijms-24-04577-f001:**
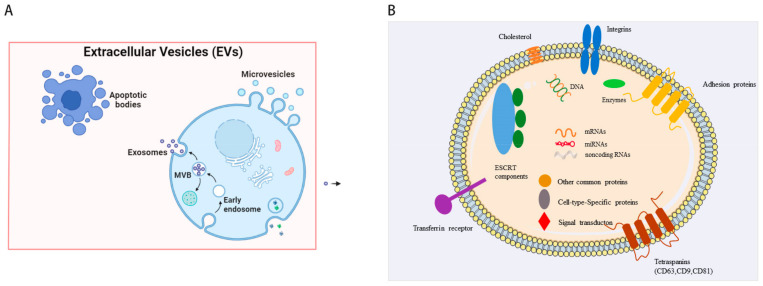
Biogenesis of EVs and schematic of exosomal molecular compositions. (**A**). EVs contain apoptotic bodies, microvesicles, and exosomes. Apoptotic bodies are formed by membrane folding, invagination, and shedding with organelles and nuclear debris. MVs are formed by the direct outward budding of plasma membranes. As for exosomes, at the very beginning, the invagination of the plasma membrane forms a cup-shaped structure termed early endosome containing cell surface proteins and other biological substances. Early endosomes then develop into late endosomes, which invaginate to form multivesicular bodies (MVBs) that finally fuse with the plasma membrane and release the exosomes. (**B**). Exosome contains various important biomarkers, such as proteins, lipids, and miRNAs.

**Figure 2 ijms-24-04577-f002:**
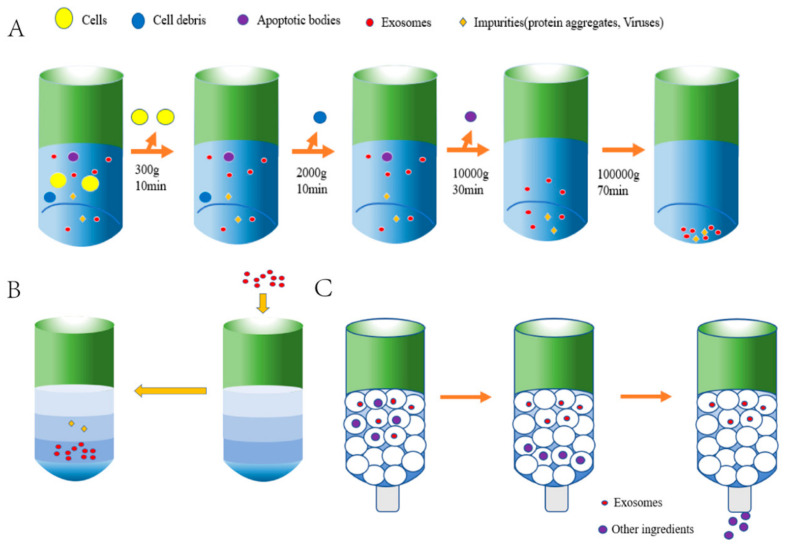
Exosome separation diagram. (**A**). Ultracentrifugation. First, the samples are centrifuged at 300× *g*, 2000× *g,* and 10,000× *g* to remove larger cells, cell debris, and dead cells. Secondly, the exosomes are isolated via ultracentrifugation twice at a speed of more than 100,000× *g*. (**B**). Isopycnic density gradient centrifugation. Impurities are firstly removed via low-speed centrifugation, and then, the separated samples are added to the constructed density medium (3%, 35%, 45%, and 90%) for separation. (**C**). Size-exclusion chromatography.

**Figure 3 ijms-24-04577-f003:**
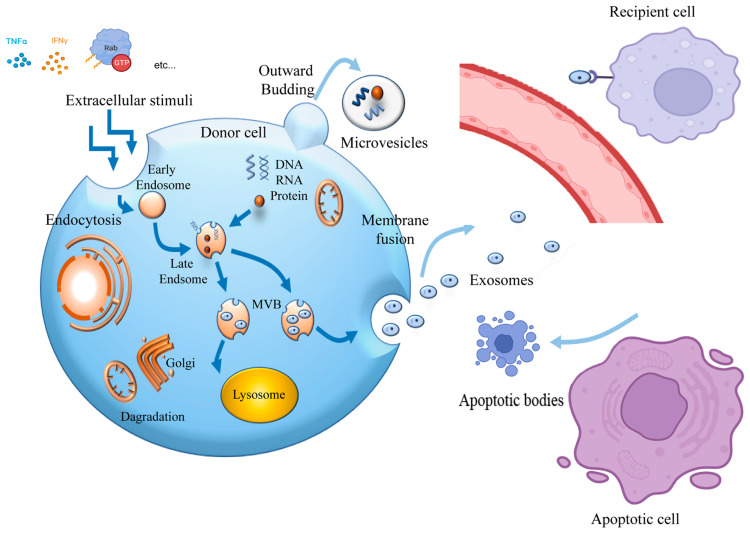
Biogenesis and information exchange of exosomes. The invagination of the plasma membrane forms a cup-shaped structure, which includes proteins on the cell surface and some components in the extracellular environment, such as proteins, lipids and metabolites, making up the early endosomes. Early endosomes then develop into late endosomes and invaginate to form intraluminal vesicles, and the cytoplasmic components also enter intraluminal vesicles, and then, late endosomes form multivesicular body. Finally, multivesicular body fuses with the plasma membrane and the exosomes are released. The receptor cells mainly interact with the exosomes through three ways: (1) the exosomes bind to the receptors on the cell membrane; (2) the exosomes fuse directly with the cell membrane to release the contents; (3) the exosomes directly enter the cytoplasm in a complete form through cellular pinocytosis or phagocytosis.

**Table 1 ijms-24-04577-t001:** The drug delivery of iPSC-sEVs in various disease models.

Cell Sources	Characterization	Models	Therapeutic Effects	Cargos	Reference
miPSCs	TEM	MI	Mitigate cardiac remodeling and improve cardiac functions post myocardial infarction		[81]
miPSCs	WB (CD63, Tsg101)	MIRI	Prevent cardiomyocyte apoptosis in ischemic myocardium	miR21, HIF-1α-regulated miR210	[82]
miPSCs	EM, FCM, RT-PCR	MIRI	Improve LV function and enhance angiogenesis	global miRNA and proteomic profiling performed	[21]
hiPSCs	FCM, BCA	MI	Reduce fibrosis in infarcted mice hearts	CD82	[83]
hiPSCs	TEM, NTA	MI	Facilitate cardiac repair through circulating miRNAs	circulating miRNAs	[84]
hiPSCs	NTA, WB (CD63)	HF	Involved in the remodeling process and observed in primary cardiomyocytes	miRNA mRNA	[85]
hiPSCs	TEM, WB (CD63, CD9)	Endothelial cell in vitro	Improve cardiac function and repair	miRNA	[86]
hiPSCs	TEM, NTA	H9c2 in vitro	Protect against oxidative-stress-induced apoptosis	miRNA	[87]
hiCMs	WB (CD63, CD81)	Dys-iCMs In vitro	Decrease reactive oxygen species and delay mitochondrial permeability		[88]
hiCMs	TEM, WB (CD63, CD81)	MI	Facilitate cardiac repair and avoid immune rejection	miRNA, LncRNA	[89]
hiCMs	WB (CD81, CD63, flotillin-1, TSTG101)	MI	Improve recovery from myocardial infarction in swine		[12]
hiMSCs	EM, NanoFCM	MI	Promote cell viability through activating the Akt/Nrf2/HO-1 axis and improve cardiac function		[90]
hiMSCs	TEM, immunoblot	Rat skin wound model	Promote collagen synthesis and angiogenesis		[91]
hiMSCs	EM, NTA	HF	Improve cardiac function and increased EF relative to baseline values	miRNA	[92]
hiMSCs	TEM, NTA, RT-PCR	Ischemic Adult Human Cardiomyocytes	Alter cardiac tissue-level remodeling	miR21-5p	[93]

## Data Availability

The data that support the findings of this study are available from the corresponding author upon reasonable request.

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
