# Peer review of "Small Extracellular Vesicles Derived from Induced Pluripotent Stem Cells in the Treatment of Myocardial Injury"

_ijms, 2023, doi:10.3390/ijms24054577_

Round 1
Reviewer 1 Report
The review by Meng et al “Extracellular vesicles derived from iPSCs in the treatment in of myocardial injury” is up to date and the Authors discuss important aspects of the iPSCs- EVs application. It may be of interest to both clinicians and researchers.
However, at the moment, some important sections of the review are superficial, while other sections can be shortened, since similar sections appear in many other reviews.
After improvement, it may be of interest to both clinicians and researchers.
I have some questions and comments that should be taken into account before publication.
1) Authors have to be more specific. As they state their Review is dedicated and they focus in the paper at the exosomes – they have to consider and talk precisely about exosomes. Otherwise, that’s confusing for readers. I would recommend to change the title in this context.
2) Paragraph 3. Contents and physiological function of iPSCs-EVs:
I didn't see anything new in this section. No specific information available.
In addition, the content of exosomes is not limited to microRNA. Also, only mentioned studies on murine iPSCs. Could Authors discuss related information on human iPSC-EVs?
Several papers already discuss some function of microRNAs for CDV in detail ( For example , Characteristics and Roles of Exosomes in Cardiovascular Disease by Yuan Zhang et al., 2017 https://doi.org/10.1089/dna.2016.3496; The therapeutic and diagnostic role of exosomes in cardiovascular diseases by Parvin Zamani et al., 2019 https://doi.org/10.1016/j.tcm.2018.10.010 )
If you have this part – it have to be deeper
3) The next section is similar to the preceding one in some ways, but it does not disclose the characteristics of the drug delivery system, such as how the selectivity of EVs is achieved. components of membranes, such as Rab proteins?
4) Paragraph 4. Table1. Line 202-208. In that context – EVs state for exosomes? Are all listed results come from the gene-edited patient-specific iPSCs? What biomarkers were used? This part is of undoubted interest and should be presented and discussed deeper. Perhaps, this can be done by the reduction of the section devoted to exosome separation methods, which is already described and presented in numerous reviews.
5) Many issues and challenges need to be resolved in order to effectively harness the therapeutic potential of exosomes for clinical applications. What are they? For the Review purpose, one line – 333 – is definitely not enough.
Although extensive research has provided valuable insights in the field of exosomes. Authors discuss biogenesis of exosomes – that’s not an exact scope of their review. Readers can find that everywhere.
6) The prospect of drug interactions is not discussed.
7) What is important to discuss - the exact mechanisms of cargo sorting, interaction between exosomal contents and exosome secretion, delivery into correct compartment - this part is important but currently missing.
Minor: Please, check the titles (5.1. MI/5.3. Coronary heart disease) to make sure they are uniformed.
Author Response
We gratefully thank for the precious time you spent making constructive remarks of our review.
1) Authors have to be more specific. As they state their Review is dedicated and they focus in the paper at the exosomes – they have to consider and talk precisely about exosomes. Otherwise, that’s confusing for readers. I would recommend to change the title in this context.
Answer: Thank you very much. We would like to change the title as “Small Extracellular Vesicles derived from induced pluripotent stem cells in the treatment of myocardial injury”. Given that latest MISEV guidelines suggest the use of “EVs” to generally denote a heterogeneous extracellular vesicle population, and “exosomes” are defined as small extracellular vesicles that are released upon the exocytosis of MVBs filled with ILVs, in this review, we have added Fig.1A to illustrate in more detail the composition and classification of extracellular vesicles, as well as the relationship between them.
2) Paragraph 3. Contents and physiological function of iPSCs-EVs:
I didn't see anything new in this section. No specific information available.
In addition, the content of exosomes is not limited to microRNA. Also, only mentioned studies on murine iPSCs. Could Authors discuss related information on human iPSC-EVs?
Several papers already discuss some function of microRNAs for CDV in detail ( For example , Characteristics and Roles of Exosomes in Cardiovascular Disease by Yuan Zhang et al., 2017 https://doi.org/10.1089/dna.2016.3496; The therapeutic and diagnostic role of exosomes in cardiovascular diseases by Parvin Zamani et al., 2019 https://doi.org/10.1016/j.tcm.2018.10.010 )
If you have this part – it have to be deeper
Answer: We have removed this part, thanks for your suggestion.
3) The next section is similar to the preceding one in some ways, but it does not disclose the characteristics of the drug delivery system, such as how the selectivity of EVs is achieved. components of membranes, such as Rab proteins?
Answer: Thank you for your suggestion. We have made a supplement to this part. For example, Rab GTP enzymes such as RAB11, RAB35, RAB27A and RAB27B participate in the production of exosomes through vesicle budding. The expression of exosome markers such as CD63 and CD81 was proved to be decreased due to the silencing of RAB27A and RAB27B.
4) Paragraph 4. Table1. Line 202-208. In that context – EVs state for exosomes? Are all listed results come from the gene-edited patient-specific iPSCs? What biomarkers were used? This part is of undoubted interest and should be presented and discussed deeper. Perhaps, this can be done by the reduction of the section devoted to exosome separation methods, which is already described and presented in numerous reviews.
Answer: Thank you very much. We have corrected it to sEVs, which means exosomes. More information can be found in the table 1.
5) Many issues and challenges need to be resolved in order to effectively harness the therapeutic potential of exosomes for clinical applications. What are they? For the Review purpose, one line – 333 – is definitely not enough.
Answer: Taking into account the suggestions of this valuable advice, we have reorganized and rewritten this section to express the challenges of exosome application more fully and specifically.
6)The prospect of drug interactions is not discussed.
Answer: We appreciate it very much for this good suggestion. We have adjusted the “Prospects and Conclusions” part to add this section.
7) What is important to discuss - the exact mechanisms of cargo sorting, interaction between exosomal contents and exosome secretion, delivery into correct compartment - this part is important but currently missing.
Answer: We thank you for your careful reading of our manuscript and providing us with keen scientific insight. This issue is supplemented in the part 3 “Drug delivery of iPSCs-sEVs in the repair of myocardial injury”.
Author Response
Dear Reviewer,
We gratefully thanks for the precious time you spent making constructive remarks of our review. We have modified our manuscript as you suggested.
Line 4; Author needs to specify the author name exactly.
Answer: Thank you for your reminder. It has been done.
Line 31; Reframe the sentence “In the past 30 years, the related mortality rate has increased by 50%”
Answer: This sentence has been reframed.
Line 36; Reframe the iPSC definition for betterment.
Answer: This has been reframed.
Line 42-43; Reference 10, needs to cross check with the statement.
Answer: It has been done.
Line 75-76; Check with the reference and cite appropriate reference.
Answer: Thank you for your reminder. It has been done.
Line 81-82; if possible, describe with the protein levels in iPSC and iPSC-EV with elaborate description.
Answer: Thank you for your valuable advice. It has been done.
Line 84-86; This sentence is repetition, already mentioned in abstract.
Answer: Thank you for your careful review. It has been corrected.
Line 105; The reference says, “Enhanced BCA protein analysis kit”.
Answer: Thank you for your careful review. It has been corrected.
Line 117; Figure 2, Abbreviation for TFF & MWCO not mentioned in whole manuscript.
Answer: We have already adjusted. Thank you very much.
Line 125-127; “and” mentioned frequently, so reframe for appropriateness.
Answer: Thank you for your guidance, we corrected this accordingly.
Line 132; Abbreviation not mentioned for “SEC”.
Answer: We have already mentioned on Line 94, when it first appeared in the review.
Line 142-150; Paragraph is fully reused from the cited article.
Line 142-150; AF4 & ANSWER method needs to elaborate more and if possible explain with diagrammatic representation for better reach in separate paragraphs.
Answer: For these two comments above, according to the comments of the first reviewer, we have adjusted “Isolation of sEVs from induced pluripotent stem cells” part, and put more discussion on the application. Thank you so much.
Line 156-157; Check for iPSCs-EV biomarkers CD9 and CD81, give appropriate reference.
Answer: Thank you for your guidance, we corrected this accordingly.
Line 158-161l Reframe the sentence to single sentence.
Answer: We have deleted here, and this part has been adjusted to the comparison of iPSCs and iPSCs-EVs in protein in the first section.
Line 163; Instead of “Moreover, among the” change to “Mendel et al reported”
Answer: Thank you for your guidance. We corrected this accordingly.
Line 168; Check this reference 66 and cite appropriately reference.
Answer: Thank you for your careful review. It has been corrected.
Line 187- 189; Add reference to this sentence.
Answer: Thank you for your careful review. The reference has been added.
Line 199; Check reference 71 and cite appropriate article.
Answer: Thank you for your careful review. It has been corrected.
Line 209; Table1. References “80, 86, 88, 98, 99, 108” Check all this reference and change appropriate reference.
Answer: Thank you for your careful review. These have been corrected.
Line 215; Figure 3, Kindly describe more about the extracellular stimuli with specific examples in the figure.
Answer: Thank you for this valuable suggestion. We have updated Fig.3 according to the comment.
Line 232; Check reference 38 and change appropriate reference.
Answer: Thank you for your careful review. It has been corrected.
Line 246; Reference article describes about miR-146a, miR-22 and miR-24, but not describes about miR-21. kindly give reference or change to appropriate miR's.
Answer: Thank you for your reminder. It has been revised.
Line 256; Check with the reference 81 is not dealt about primary rat CMs, cite appropriate reference.
Answer: Thank you for your careful review. These have been corrected.
Line 300; Check with the reference 106, cite appropriate reference.
Answer: Thank you for your careful review. It has been corrected.
Reviewer 3 Report
In this review article, Meng et al highlight the importance of extracellular vesicles (EV) derived from iPSCs as an innovative therapeutic agent to treat myocardial injury. The authors also nicely explained the techniques for isolating each type of vesicle along with its limitations. They also highlight the different successes of using iPSC-derived EVs. Briefly, they also describe the limitation of using those EVs and possible solutions for that. However, there are a few minor parts, which need to revise to improve the quality of the manuscript.
a) In the author ship area, the author forgot to put the name of 3rd author. Please add the name of 3rd author after 'and'.
b) Please provide a table to summarize the utilization of iPSC as well as iPSC-derived EV as a therapeutic agent and its outcome (if available).
c) In section 6 “Challenges in the treatment of CVD with iPSCs-EV”, the authors largely describe minor limitations. There are several major limitations to treating iPSC-derived EVs. Please add those major limitations and critically discuss them in the section with possible alternative approaches.
For example, from recent studies, it is clear that EVs could cross the blood-brain barrier results neurological disorders like stress, anxiety, depression etc (https://doi.org/10.1038/s41398-019-0459-9). Moreover, Stress is the most critical factor, which could induce myocardial disease as well as tumorigenesis and promote cancer development via modulation of genetic and epigenetic changes. Moreover, it could modulate the therapeutic efficacy via the regulation of differential gene expressions (https://doi.org/10.3389/fonc.2020.01492; https://doi.org/10.1038/s41568-021-00395-5). So stress or other physiological condition will largely influence the effect of iPSC treatment.
On the other hand, lots of recent research already came out showing that stress management via meditation related to mind body and breath could help to get better outcomes of the treatment. (https://doi.org/10.1073/pnas.2110455118; https://doi.org/10.1186/s13063-018-3103-8; doi: 10.3389/fpsyg.2021.635816; DOI 10.1007/s11764-012-0252-8, https://doi.org/10.1016/j.urolonc.2020.09.011; doi:10.1002/nur.22169 ).
So please revise section 6 considering the above-mentioned limitation and discuss if the iPSCs-EV in combination with stress management meditation could be fruitful as an alternative approach.
Author Response
Thank you so much four your comments and professional advice concerning our manuscript. These opinions help to improve academic and rigor of our article.
a) In the author ship area, the author forgot to put the name of 3rd Please add the name of 3rdauthor after 'and'.
Answer: Sorry for this error, we have corrected it.
b) Please provide a table to summarize the utilization of iPSC as well as iPSC-derived EV as a therapeutic agent and its outcome (if available).
Answer: Thank you for your rigorous consideration. We combined the comments of the three reviewers and revised Table 1 to illustrate the therapeutic strategies for iPSCs-sEVs.
c) In section 6 “Challenges in the treatment of CVD with iPSCs-EV”, the authors largely describe minor limitations. There are several major limitations to treating iPSC-derived EVs. Please add those major limitations and critically discuss them in the section with possible alternative approaches.
For example, from recent studies, it is clear that EVs could cross the blood-brain barrier results neurological disorders like stress, anxiety, depression etc (https://doi.org/10.1038/s41398-019-0459-9). Moreover, Stress is the most critical factor, which could induce myocardial disease as well as tumorigenesis and promote cancer development via modulation of genetic and epigenetic changes. Moreover, it could modulate the therapeutic efficacy via the regulation of differential gene expressions (https://doi.org/10.3389/fonc.2020.01492; https://doi.org/10.1038/s41568-021-00395-5). So stress or other physiological condition will largely influence the effect of iPSC treatment.
On the other hand, lots of recent research already came out showing that stress management via meditation related to mind body and breath could help to get better outcomes of the treatment. (https://doi.org/10.1073/pnas.2110455118; https://doi.org/10.1186/s13063-018-3103-8; doi: 10.3389/fpsyg.2021.635816; DOI 10.1007/s11764-012-0252-8, https://doi.org/10.1016/j.urolonc.2020.09.011; doi:10.1002/nur.22169 ).
So please revise section 6 considering the above-mentioned limitation and discuss if the iPSCs-EV in combination with stress management meditation could be fruitful as an alternative approach.
Answer: We are appreciative of your suggestion and modified accordingly in revised manuscript.
Round 2
Reviewer 1 Report
THe paper is improved and can be accepted for publication
Author Response
Dear reviewer,
Thank you for your positive comment.
Author Response
Dear Reviewer,
Thank you so much four your comments and professional advice concerning our manuscript. These opinions help to improve academic and rigor of our article. We have read the comments carefully and have made revision which marked in green in the paper.
Line 85; Reference 25 needs to check, not matching with the content.
Answer: Thanks for pointing this out. We have rewritten the sentence and cited the appropriate references. Thank you very much.
Line 139; In version 1, the section 3 states about the "Contents and physiological functions of iPSCs-EVs" which was merged few sentence with section 4. So modify the section title accordingly.
Answer: Thank you for your comments. In the revision of round1, the third section has been deleted and the duplication is no longer present.
Line 144-147; Whole sentence is plagiarism from reference no 61.
Answer: We are so sorry to have such a problem. we rewrote this sentence, adjusted the original text at the position of line 167, and integrated it with the repeated part of line 144-147, so that the content of this part is more clearly structured and the sentence is more smooth.
Line 151; Reference no. 65 & 66 didn’t discuss about exosomal marker CD81. Check and add appropriate reference.
Answer: We have corrected this sentence. In the article on Rab27a and Rab27b, the exosomal marker often verified is CD63.
Line 151-154; Ambiguous sentence, required clear statement of previous work.
Answer: We have revised this part to describe more clearly the purpose and significance of citing articles.
Line 327-335; This paragraph is deviated from the article objective. If the meditation improves EV production needs to discuss above the article with references and cohort studies.
Answer: We really thank the reviewer for his/her careful work. Considering that it is not closely related to the above, we have deleted this section.